

# Reviews and syntheses: Insights into deep-sea food webs and global environmental gradients revealed by stable isotopes (δ15N, δ13C) and fatty acids trophic biomarkers

Camilla Parzanini[1], Christopher C. Parrish[1], Jean-François Hamel[2], Annie Mercier[1]

[1]Department of Ocean Sciences, Memorial University, St. John's, NL, Canada

[2]Society for Exploration and Valuing of the Environment (SEVE), St. Philips, NL, Canada

**Correspondence to:** Camilla Parzanini (cparzanini@ryerson.ca). Current address: Department of Chemistry and Biology, Ryerson University, Toronto, ON, Canada

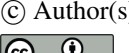



**Abstract.** Biochemical markers developed initially for food-web studies of terrestrial and shallow-
water environments have only recently been applied to deep-sea ecosystems (i.e. in the early
2000s). For the first time since their implementation, this review took a close look at the existing
literature in the field of deep-sea trophic ecology to synthesize current knowledge. Furthermore, it
provided an opportunity for a preliminary analysis of global geographic (i.e. latitudinal, along a
depth gradient) trends in the isotopic ($\delta^{15}$N, $\delta^{13}$C) and fatty acid composition of deep-sea taxa.
Results revealed significant relationships along the latitudinal and bathymetric gradients. Deep-sea
animals sampled at temperate and polar latitudes displayed lower isotopic ratios and greater
proportions of essential ω3 long-chain polyunsaturated fatty acids (LC-PUFA) than did tropical
counterparts. Furthermore, $\delta^{15}$N and $\delta^{13}$C ratios as well as proportions of arachidonic acid
increased with increasing depth. Since similar latitudinal trends in the isotopic and fatty acid
composition were found in surface water phytoplankton and particulate organic matter, these
results highlight the link across latitudes between surface primary production and deep-water
communities. Because global climate change may affect quantity and quality (e.g. levels of
essential ω3 PUFA) of surface primary productivity, and by extension those of its downward flux,
the dietary intake of deep-sea organisms may likely be altered. In addition, because essential ω3
PUFA play a major role in the response to temperature variations, climate change may interfere
with the ability of deep-sea species to cope with potential temperature shifts. Importantly,
methodological disparities were highlighted that prevented in-depth analyses, indicating that further
studies should be conducted using standardized methods in order to generate more reliable global
predictions.



## 1 Introduction

### 1.1 Historical background of biochemical biomarkers in deep-sea food-web studies

While the use of biochemical biomarkers in marine food-web studies has a long and successful tradition in shallow-water ecosystems, starting from the 1970s with the use of stable isotopes (McConnaughey and McRoy, 1979) and lipids (Lee et al., 1971), their application in deep-water environments is relatively new (e.g. Iken et al., 2001; Polunin et al., 2001; Howell et al., 2003). Undoubtedly, technological advances made over the past few decades have allowed the exploration of ever deeper ecosystems with more refined techniques. Iken et al. (2001) were among the first to provide the analysis of a deep-sea food web, which was sampled at a depth of ~4840 m at the Porcupine Abyssal Plain (PAP, Northeast Atlantic), by using bulk stable N and C isotope ratios ($\delta^{15}N$ and $\delta^{13}C$ respectively) as trophic markers. In the same year, Polunin et al. (2001) used the same approach to study the trophic relationships of a slope megafaunal assemblage collected off the Balearic Islands (western Mediterranean). Since these first two investigations, several others have been carried out across different oceanic regions and climes, such as the Canadian Arctic (Iken et al., 2005), the Arabian Sea (Jeffreys et al., 2009), and the Sea of Japan (Kharlamenko et al., 2013). Furthermore, over the past decade, it has become evident that the simultaneous use of different trophic markers (e.g. $\delta^{15}N$, $\delta^{13}C$, and fatty acids, FA) and techniques (e.g. bulk or compound specific isotope analysis, as well as FA, gut content and morphometric analyses) provides a more complete picture of trophic structure and dynamics. Indeed, while the first investigations relied on a single method (Iken et al., 2001; Polunin et al., 2001; Howell et al., 2003), the latest trend in deep-sea food-web studies favours an integrative approach, which maximizes the efficiency of each technique, while increasing the resolution of the investigation (e.g. Stowasser et al., 2009; Parzanini et al., 2017).

For the first time since the implementation of trophic markers in studies of deep-sea food webs two decades ago, this review synthesizes current knowledge in this growing field of research.





In addition, it provides a preliminary overview of large-scale geographic trends from the analysis of
isotopic and FA data, along with guidance for future investigations. In particular, the present
contribution i) briefly defines various trophic biomarkers and their respective advantages; ii)
describes deep-sea food webs, based on examples from the literature; iii) lists the sources of
variation among the different studies to highlight pitfalls and gaps; and iv) provides a preliminary
quantitative analysis across studies by using relevant datasets.
**1.2 Comparison of major trophic markers**
The analysis of gut contents was among the first techniques (together with *in situ* observation of
feeding behaviors) applied in trophic ecology and food-web studies in aquatic systems (Gartner et
al., 1997; Michener and Kaufman, 2007). Subsequently, other methods were developed as
alternative or supplementary means of studying diet and feeding behaviors within the same
ecosystems. Among them, the use of biochemical markers as trophic tracers rapidly grew in
popularity in food-web ecology, since it is relatively simple and should overcome many of the issues
ascribed to gut content analysis (Michener and Kaufman, 2007). In this regard, Table 1 lists
strengths and drawbacks of gut content analysis and of the two most popular biochemical
techniques, i.e. bulk stable isotope and FA analyses. For instance, bulk stable isotope and FA
analyses may, theoretically, be performed on any species, regardless of feeding mode and food
sources, whereas gut content analysis can only be applied to those organisms characterized by a
sufficiently large and full stomach. Except in cases where individuals are too small and have to be
analyzed whole, biochemical analyses are typically conducted on target tissues (e.g. muscle) that
provide long-term dietary data and reduce intra-individual variability (Table 1). In addition, the use
of biochemical tracers requires shorter processing times than gut content analysis. Thanks to this
integrative approach and faster output, the application of food-web tracers has been particularly
helpful in deep-sea studies, which are often plagued by financial and logistical constraints.
Furthermore, due its relative ease of use, it has favoured the analysis of wider sets of taxa/feeding
guilds, primary producers included, rather than focusing on one or a few focal groups. However, the



interpretation of isotopic and FA data is complex, and both techniques require dedicated and
sophisticated instrumentation (e.g. gas chromatograph, mass spectrometer) and knowledge of
intrinsic sources of variations (see Sect. 1.4). Although each method needs a sufficient sample
size, only gut content analysis may provide direct and clear evidence of the diet (Table 1).
Therefore, as stated above, the latest trend in trophic ecology advocates a multifaceted approach,
on the understanding that each technique may offer unique and valuable data.

The principle behind the use of food-web tracers is that the biochemical signature of

consumers reflects that of their diet. Among them, $\delta^{15}N$ and $\delta^{13}C$ are the most popular. While the
former is used to study trophic positions and dietary sources, with an enrichment factor of 2-4‰
between a consumer and its food (Minagawa and Wada, 1984); the latter undergoes little
fractionation (<1‰) and, therefore, is used to distinguish primary food sources (McConnaughey
and McRoy, 1979). For further details, refer to Sulzman (2007) and Michener and Kaufman (2007)
who have provided extensive reviews on the chemistry behind stable isotopes and their use as
food-web tracers, respectively. In addition, sterols, FA and amino acids, which are important
constituents of lipids (for the former two) and proteins (for the latter), have successfully been used
to study trophic relationships and dietary sources in deep-water systems (Howell et al., 2003;
Drazen et al. 2008a, 2008b). Their use is based on the principle that certain FA and amino acids
are considered essential for animals, being required for optimal fitness. However, most species
cannot synthesize these essential compounds *de novo* and, therefore, they must gain them through
their diet. Indeed, only primary producers and a few consumers possess the enzymatic apparatus
to synthesize essential FA and amino acids *de novo*. Conversely, a few taxa are unable to
synthesize sterols *de novo*, which are critical for them; therefore, they have to acquire these
essential sterols through diet (Martin-Creuzburg and Von Elert, 2009). Because sterols, FA, and
amino acids undergo little or no alteration when consumed, it is possible to detect dietary sources
within the consumers' tissues (Parrish et al., 2000). The isotopic signature of amino acids can also
be used to study trophic position through compound specific analysis ($\delta^{15}N$), as some of these
acids show trophic enrichment (Bradley et al., 2015). Detailed information about FA analysis was



outside the scope of this study, and is provided by Parrish (2009) and Iverson (2009); whereas the
use of sterols as food-web tracers was outlined in Martin-Creuzburg and Von Elert (2009) and
Parrish et al. (2000). McClelland and Montoya (2002) and Larsen et al. (2009), conversely, discuss
the use of amino acids as trophic biomarkers.
**1.3 Understanding deep-sea food webs through biochemical markers**
As there is no photosynthetically-derived primary production in the deep sea, deep-water
ecosystems are mostly heterotrophic (Gage, 2003), and may hence largely rely on particulate
organic matter (POM) that passively sinks from the surface waters as a primary source of nutrients
(Hudson et al., 2004). Nonetheless, food can also be actively transported down by those animals
that carry out vertical diel migrations through the water column (Trueman et al., 2014); it can also
be provided by the occasional fall of large animal carcasses (Smith and Baco, 2003); and/or by
lateral inputs, from inland and shelf areas towards abyssal offshore regions (Pfannkuche, 2005).
Although most of the deep-water ecosystems are heterotrophic, a few, such as hydrothermal vents
and cold seeps, are fuelled by chemical energy (e.g. methane, hydrogen sulfide) and rely on
chemosynthetic microorganisms for the production of organic matter. Each of these primary food
sources has a specific isotopic composition and biochemical signature, resulting from a
combination of chemical and physical processes reflective of its origin. By knowing the composition
of the food source(s) that fuel(s) a given food web, it is possible to re-construct its trophic structure
and dynamics. Conversely, by measuring the signatures of the food-web components, it is possible
to assess food sources on which they rely. For instance, Iken et al. (2001) showed that
phytodetritus was the primary energy input of the deep-sea benthic community at PAP, and also
defined two different trophic pathways: a pelagic and isotopically lighter one in which sinking POM
and small pelagic prey constituted the main food sources; and a benthic and more isotopically
enriched trophic pathway, fuelled by degraded sedimented POM. In fact, once POM settles on the
seafloor, it undergoes continuous degradation by microbes and is reworked through bioturbation
and feeding activities, thus leading to a more isotopically enriched material relative to the sinking



one (Iken et al., 2001). Depending on the primary food source they relied on, benthic organisms at
PAP were thus characterized by either lower or higher values of δ¹⁵N. Similar scenarios of dual
trophic pathways characterizing benthic systems were also found by Iken et al. (2005) in the
Canadian Arctic; Drazen et al. (2008b) in the North Pacific; Reid et al. (2012) within the benthic
community sampled on the mid-Atlantic Ridge; Valls et al. (2014) in the western Mediterranean;
and Parzanini et al. (2017) in the Northwest Atlantic. Moreover, Kharlamenko et al. (2013) used
both stable isotopes and FA to study the dietary sources of benthic invertebrates collected along
the continental slope (500-1600 m depth) in the Sea of Japan. The authors recognized different
trophic pathways (i.e. planktonic, benthic, microbial) and dietary sources by using biochemical
tracers; and they proposed a strong link with the primary production of the surface waters, as the
FA composition of the deep-sea echinoderms and mollusks was similar to that of the shallow-water
counterparts.

As POM sinks through the water column, its δ¹⁵N increases, reflecting the preferential

assimilation of the lighter isotope, ¹⁴N by microbes; in particular, a gradient in POM δ¹⁵N has been
detected with depth, where POM at greater depths is more enriched (Altabet et al., 1999). For this
reason, Mintenbeck et al. (2007) carried out a study in the high-Antarctic Weddell Sea to assess
whether this gradient was reflected in the isotopic signature of POM consumers sampled at 50-
1600 m. In this regard, only those organisms feeding directly on sinking POM (e.g. suspension
feeders) showed increasing values of δ¹⁵N with depth, whereas the increase was less evident for
the deposit feeders (Mintenbeck et al., 2007). Similar results for suspension feeders were obtained
by Bergmann et al. (2009) who analyzed a benthic food web sampled at the deep-water
observatory HAUSGARTEN, west of Svalbard (Arctic), between 1300 and 5600 m depth.
Conversely, deposit feeders exhibited a negative trend along the bathymetric gradient in terms of
δ¹⁵N, and predator/scavengers were not affected. In another study, Sherwood et al. (2008) did not
detect any relationships with depth in the δ¹⁵N values measured from cold-water corals collected on
a slope environment in the Northwest Atlantic. Among the explanations suggested for these
inconsistencies and differences among feeding groups, Mintenbeck et al. (2007) and Sherwood et



al. (2008) included feeding preferences with respect to the size and sinking velocity of POM.
According to these authors, only those organisms feeding on small particles of sinking POM should
reflect a bathymetric gradient in $\delta^{15}$N. In fact, small-sized particles sink at a lower velocity and,
therefore, experience high rates of degradation, with more evident changes in $\delta^{15}$N (Mintenbeck et
al., 2007). Based on these findings, depth-stratified sampling should ideally be conducted when
studying a system characterized by a bathymetric gradient, as it would prevent biases in the
interpretation of the isotopic data.

Deep-water systems are generally characterized by a limited food supply, as the quantity of

food being transferred from the surface to the bottom diminishes with increasing depth (Gage,
2003). In addition, in temperate areas, food arrives as intermittent pulses, following the spring and
late summer blooms of primary (and secondary) productivity. For this reason, deep-water benthic
communities can only rely on fresh, high-quality phytodetritus within short temporal windows
following algal blooms; whereas reworked and resuspended POM fuels these communities for the
rest of the year (Lampitt, 1985). Deep-sea benthic organisms have hence developed adaptations
and strategies to increase their feeding success and minimize competition for food, including
trophic niche expansion and specialization. In this regard, certain benthic taxa (e.g. pennatulacean
corals, hexactinellid sponges) and/or feeding groups (e.g. suspension and deposit feeders) at PAP
showed vertical extension of their trophic niches (i.e. omnivory) which, according to Iken et al.
(2001), was most likely driven by a strong competition for food. In other words, some species
belonging to the same taxon or feeding guild shared similar food sources (i.e. exhibiting similar
$\delta^{13}$C values), but they were located at different trophic levels (i.e. exhibiting a wide range of $\delta^{15}$N).
Similarly, Jeffreys et al. (2009) reported trophic niche expansion among and within feeding guilds
sampled between 140 and 1400 m depth, at the Pakistan margin (Arabian Sea). Pennatulacean
corals and other sestonivorous cnidarians, for example, displayed the greatest niche expansion;
they fed not only on POM, but also on small invertebrates (e.g. zooplankton). Moreover, ophiuroids,
which are typically selective deposit feeders, switched to an omnivorous diet under food-limited
conditions (Jeffreys et al., 2009). Apart from trophic niche expansion, Iken et al. (2001) proposed



that specialization on certain food items represented another adaptation developed by benthic
organisms at PAP to mitigate competition for food. Holothuroid echinoderms, for instance, were
thought to accomplish food specialization through a combination of different factors involving
changes in morphology, mobility, and digestive abilities (Iken et al., 2001). Further examples of
trophic niche segregation and food partitioning, as strategies to minimize competition, were also
reported for deep-sea demersal fishes in the Northwest Mediterranean Sea (Papiol et al., 2013)
and for asteroid echinoderms in the Northwest Atlantic (Gale et al., 2013). Howell et al. (2003)
detected trophic niche expansion across different species of deep-sea asteroids (1053-4840 m) by
analyzing their FA composition. In particular, multivariate analysis of FA proportions discriminated
three different feeding guilds among the asteroids analysed, including mud ingesters,
predators/scavengers, and suspension feeders.
**1.4 Sources of variation across studies**
When comparing studies relying on biochemical analysis, there are numerous sources of variation,
which may influence results and findings, and also prevent the detection of similarities and general
trends. However, their importance may depend on the scale of the investigation (i.e. local, regional,
or global). In this section, the main sources of variation are illustrated and explained by type (Table

2).

**1.4.1 Biological sources**
Age, size, and sex, whether related to diet, determine natural intraspecific variability in the isotopic
and FA compositions of organisms, which may affect data interpretation of small spatial scale
investigations. At a basic level, sessile and sedentary taxa typically experience a transition from a
pelagic to a benthic lifestyle between the larval and the juvenile stage (Rieger, 1994). Research has
also shown that certain deep-sea fish experience changes in diet with age, typically with younger
individuals preying upon benthic organisms and adults feeding on prey that are larger and of
benthopelagic origin (Mauchline and Gordon, 1984; Eliassen and Jobling, 1985). Stowasser et al.



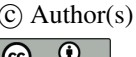

(2009) combined stable isotope analysis (SIA) and FA analysis to detect ontogenetic shifts in the
diet of the fish *Coryphaenoides armatus* and *Antimora rostrata*, collected at depths between 785
and 4814 m at PAP (Northeast Atlantic). By looking at their biochemical composition, the two
species switched from active predation to scavenging with increasing size. Similar results are
reported in Drazen et al. (2008c) for macrourid fish species from the eastern North Pacific.
Conversely, although Reid et al. (2013) detected size-related trends in the $\delta^{13}C$ of deep-water fish
collected from the Mid-Atlantic Ridge at 2400-2750 m depth, the authors were not able to
distinguish whether these results were due to ontogenetic changes in diet or merely to an effect of
increasing size, within the size-range sampled. Moreover, $\delta^{15}N$ and trophic position may increase
with body size in adult shallow-water fish, as larger predatory fish ingest larger, more isotopically
enriched prey (Badalamenti et al., 2002; Galván et al., 2010).

The potential influence of sex as a source of variation in biomarker studies has not received

as much attention and remains ambiguous. Nonetheless, Boyle et al. (2012) studied whether diet
and trophic position varied between sexes in deep-sea fish species collected at 55 -1280 m depth
in the eastern North Pacific using gut content and stable isotope analysis of muscle tissue. The
authors did not detect any difference between sexes, but variations in trophic position were
encountered when analyzing fish of different sizes (Boyle et al., 2012). An investigation of the
oceanic squid *Todarodes filippovae* sampled within a depth range of 13-380 m in the southwestern
Indian Ocean by Cherel et al. (2009), revealed that females had higher values of $\delta^{15}N$, and thus
occupied a higher trophic position. However, because *T. filippovae* exhibits sexual dimorphism in
body size, this difference was ultimately shown to be driven by size, i.e. no $\delta^{15}N$-variations were
detected when females and males of similar sizes were compared (Cherel et al., 2009). Sex may
constitute a source of variation in relation to diet in those species that exhibit extreme cases of
sexual dimorphism, as in deep-sea anglerfish (Shine, 1989). However, investigation of the role of
sex on intraspecific variability will need to be carried out across a broader taxonomic scope before
drawing generalizations.



### 1.4.2 Environmental sources


Larger-scale (e.g. regional, global) comparative studies among deep-sea habitats are complicated
by the wide bathymetric ranges they may occupy, anywhere between 200 and ~11 000 m depth.
Depth may constitute a major driver of variation of $\delta^{15}$N and $\delta^{13}$C in deep-sea organisms for two
main reasons. First, as mentioned earlier, biodegradation processes occurring within the water
column may favour the enrichment of POM as it sinks, thus influencing the stable isotope
composition of those organisms that directly feed on it (Mintenbeck et al., 2007; Bergmann et al.,
2009). Second, size-based trends and shifts in diet, hence in the isotopic composition, with depth
have been reported for deep-sea demersal fish (Collins et al., 2005; Mindel et al., 2016a, 2016b).
Likewise, deep-sea species may exhibit different lipid and FA compositions along a bathymetric
gradient, reflecting physiological adaptations to changing temperature and pressure with depth
(Parzanini et al., 2018b).
Geographic location (e.g. latitude) and season, linked to level/type of surface primary
production, nitrogen supply dynamics, as well as temperature, are also important factors to
consider when comparing studies, as large-scale temporal and spatial differences may be detected
in the organisms' isotopic composition. Stowasser et al. (2009), for instance, combined stable
isotope and FA acid analyses to study seasonal variations in the diet of 5 species of demersal fish
collected between 785 and 4814 m in the Northeast Atlantic. The authors found overall that stable
isotope and FA composition of fish varied temporally, and that these differences most likely
reflected timing and strength of food inputs sinking from surface waters. However, not all the
species (e.g. *Coryphaenoides armatus*) exhibited a strong seasonality in their biochemical
composition, probably due to the high trophic position of the species and the length of the food web
analyzed obscuring the effects of the seasonal POM inputs (Stowasser et al., 2009). Colombo et al.
(2016) detected a latitudinal gradient in the FA composition of marine species, with higher levels of
ω3-polyunsaturated fatty acids in organisms collected at polar and temperate regions in
comparison to tropical ones. Large-scale geographic effects will be further explored below, in the





exploratory analytical section; however, Fig. 1 shows where food-web studies accomplished via
biochemical tracers have been carried out, highlighting important geographic heterogeneity,
especially the limited number of investigations in the southern hemisphere.

**1.4.3 Analytical sources**

Several aspects of the SIA methodology can generate variability among studies, including type(s)
of tissue chosen for analysis, as well as sample treatment and storage, thus influencing
interpretation of small-scale investigations. For instance, lipids have lower $^{13}$C in comparison to
proteins and carbohydrates (DeNiro and Epstein, 1977), lipid-rich tissues hence display lower $\delta^{13}$C
values. In addition, there are tissues, such as liver in fish and gonads in other taxa, which are
characterized by higher turnover rates of lipids than others (e.g. white muscle), and hence
incorporate information only on the recent diet. To avoid biases caused by the presence of lipids in
tissues, several approaches may be used. Stowasser et al. (2009) and Boyle et al. (2012), for
example, opted to extract lipid from the tissues prior to analysis, whereas Sherwood et al. (2008),
Fanelli et al. (2011a, 2011b) and Papiol et al. (2013) applied a mathematical correction to their $\delta^{13}$C
data, based on the elemental C to N ratio (C:N) characterizing the samples. Other authors, such as
Polunin et al. (2001) and Carlier et al. (2009), did not apply any treatment. In the case of
mathematical corrections, two equations are currently used for deep-sea organisms, those
proposed by Post et al. (2007) and Hoffman and Sutton (2010). Since lipid extraction increases
values of $\delta^{15}$N in deep-sea fish muscle tissue (Hoffman and Sutton, 2010), this practice is not
recommended. Conversely, mathematical corrections seem to be preferable when dealing with
lipids, and they have already been applied in several studies, including those mentioned above.

Some marine organisms, such as corals and echinoderms, contain carbonate skeletal

elements. Since inorganic carbonate has higher $\delta^{13}$C values than other fractions (Pinnegar and
Polunin, 1999), it is a widespread practice to acidify these types of samples. Variations occur when
acidification is executed on samples that are simultaneously run for $\delta^{15}$N and $\delta^{13}$C, as the treatment
may affect $\delta^{15}$N data (Bunn et al., 1995). Whenever feasible, depending on both financial



constraints and the sizes of the organisms, processing samples separately for each isotope would
therefore be advisable, as in Carlier et al. (2009), Sherwood et al. (2008), and Papiol et al. (2013).

The tissues of elasmobranchs (e.g. sharks, rays) contain urea and trimethylamine oxide,

which are both [15]N-depleted; therefore, their presence may affect stable isotope data (Hussey et
al., 2012; Kim and Koch, 2012; Churchill et al., 2015). As for the inorganic carbonate issue, there is
no agreement among studies. Nonetheless, the removal of urea prior to analysis or the use of
arithmetic corrections are among the most common solutions applied to deal with the presence of
these compounds. In addition, the former seems to be the more commonly recommended and
performed, as the application of mathematical corrections requires the calculation of species-
specific discrimination factors, which is not always feasible (Hussey et al., 2012).

Sample storage is also crucial to obtain reliable data, since non-optimal preservation

methods may compromise the outcome of the investigation. Regarding the storage temperature,
while biological samples for gut content and stable isotope analysis are commonly frozen at -20°C,
if not processed soon after their collection; those for lipid analysis are either stored at -80°C
(recommended) or at -20°C prior to further processing in the lab. Since storage at -20°C might not
completely prevent lipid degradation, especially if samples are analyzed after several years, rapid
initial processing of samples and vacuum packing may reduce potential issues when freezing at -
80°C is not logistically feasible. In addition, freezing is highly recommended over chemical storage
for stable isotope analysis, as there is evidence that formalin/ethanol considerably alters the
isotopic ratios in biological tissues (Arrington and Winemiller, 2002; Syväranta et al., 2011; Xu et
al., 2011).

## 304    2 Preliminary comparative analysis

The study of large-scale trends in biological variables (e.g. distribution, biochemical composition,
biodiversity) may not only help understand general functioning and structure of ecosystems, but it
may also allow us to make predictions and support conservation initiatives. While several studies
already exist on large-scale distribution and biodiversity patterns of deep-sea species (Rex et al.,



1993; Stuart et al., 2003; Ramirez-Llodra et al., 2010), a similar approach has yet to be applied to
trophodynamics. This preliminary analysis detected global spatial trends (i.e. along latitudinal and
depth gradients) in the isotopic and FA composition of deep-water animals for the first time since
the application of biochemical tracers to the study of trophic ecology in the deep sea.

Latitudinal gradients have been detected in $\delta^{13}$C of plankton and POM collected from

surface waters in both the southern and northern hemispheres, with decreasing values towards the
polar regions (Sackett et al., 1965; Rau et al. 1982; Francois et al., 1993). Both environmental (e.g.
temperature, nutrient supply) and biological (e.g. plankton metabolism) factors have been proposed
to explain such trends (Rau et al., 1982; Francois et al., 1993). The stable N isotope signature of
surface primary production may also vary regionally, depending on the nutrient (mainly N) supply to
the phytoplankton, as well as its community structure and cell size (Choy et al., 2015; Hetherington
et al., 2017). Oligotrophic areas, characterized by marked oxygen minimum zones and by high
denitrification rates, such as the eastern tropical Pacific Ocean, typically have higher $\delta^{15}$N values
(Hetherington et al., 2017). In addition, latitudinal trends have been detected in the FA composition
of marine organisms, which tend to have higher levels of essential $\omega 3$ long-chain polyunsaturated
fatty acids (LC-PUFA) in the polar and temperate regions in comparison to the tropical ones
(Colombo et al., 2016). As POM is the main food source of most deep-sea food webs (Gage, 2003;
Hudson et al., 2004), we hypothesized that a) similar latitudinal gradients exist in the isotopic and
essential PUFA composition of deep-water organisms; and that b) the strength of these trends
varied among organisms from different habitats, i.e. pelagic, demersal, and benthic, as diversely
dependant on POM. Furthermore, as both isotopic and lipid composition of POM and as deep-sea
taxa varied along a depth gradient in the deep North Pacific (Lewis, 1967; Altabet et al., 1999),
North Atlantic (Polunin et al., 2001; Parzanini et al., 2018a, 2018b, 2017) and Arctic Ocean
(Bergmann et al., 2008), we hypothesized that similar trends could be extended to the global scale.



## 2.1 Materials and methods

### 2.1.1 Data set

This analysis focused on studies that used either bulk stable isotope or FA analysis, or a combination of them, to infer trophic relationships of deep-water organisms, as well as to study deep-sea food webs. Studies on chemosynthetic habitats (e.g. hydrothermal vents) were excluded *a priori* to avoid possible biases. In fact, these habitats are fuelled by primary dietary sources, e.g. methane, whose isotopic and FA composition is substantially different than that of POM (Rau and Hedges, 1979; Saito and Osako, 2007). Table 3 outlines the full data set collated for the present analysis, which includes 45 different studies. The literature search was carried out through Scopus and Google Scholar portals using the following key words: stable isotopes, fatty acids, food webs, deep sea, trophic ecology, and trophic relationships. These studies were used to analyze global trends in $\delta^{15}N$, $\delta^{13}C$, and the essential arachidonic (ARA, 20:4ω6), eicosapentaenoic (EPA; 20:5ω3) and docosahexaenoic (DHA, 22:6ω3) acids across deep-water communities. ARA, EPA, and DHA are the most important nutrients in aquatic ecosystems, required by organisms for optimal health (Parrish 2009), as well as excellent trophic biomarkers. In fact, whereas EPA and DHA are typically used as biomarkers in diatoms and dinoflagellates respectively (Parrish, 2013), in the deep sea, ARA is associated with microorganisms from the sediment (Howell et al. 2003). Our study focused on these three FA since they are present in all the organisms under analysis.

### 2.1.2 Variables considered

Each species from each investigation was sorted by latitude (i.e. tropical, 0 - 30°; temperate, 30 - 60°; and polar, 60 - 90°), habitat (i.e. pelagic, demersal, and benthic) depth at collection (i.e. mesopelagic, 200 – 1000 m; bathypelagic, 1000 – 4000 m; and abyssopelagic, >4000 m, whether pelagic species; bathyal 200 - 4000 m; and abyssal, >4000 m, whether benthic species), and phylum (i.e. Annelida, Arthropoda, Brachiopoda, Chordata, Cnidaria, Echinodermata, Mollusca, Nematoda, Porifera, and Sipuncula). Information about species habitat was either obtained through



WoRMS and FishBase online databases or was already included in the source paper. In addition,
species were labelled as "meso-bathypelagic" and "bathyal-abyssal", if the depth at collection was
not specified further, but the whole set of samples for a study was collected within those zones. In
the current analysis, tissue type, acidification treatment, sampling season, sex, and age were not
considered as variables, because i) they were assumed to not play a major role in global-scale
investigations and/or ii) this information was not always provided. In addition, tests were performed
on lipid-corrected and uncorrected $\delta^{13}C$ data pooled together. For analyses regarding stable
isotope composition ($\delta^{15}N$, $\delta^{13}C$), from polar to tropical regions, data were obtained from Iken et al.
(2005), Bergmann et al. (2009), van Øevelen et al. (2018); Iken et al. (2001), Madurell et al. (2008),
Sherwood et al. (2008), Carlier et al. (2009), Fanelli et al. (2009), Stowasser et al. (2009), Fanelli et
al. (2011a, 2011b), Boyle et al. (2012), Reid et al. (2012), Fanelli et al. (2013), Gale et al. (2013),
Kharlamenko et al. (2013), Papiol et al. (2013), Reid et al. (2013), Tecchio et al. (2013), Trueman et
al. (2014), Valls et al. (2014a, 2014b), Kopp et al. (2018), Parzanini et al. (2017), Preciado et al.
(2017), Parzanini et al. (2018a); Jeffreys et al. (2009), Churchill et al. (2015), and Shipley et al.
(2017) (Table S1). FA composition (ARA, EPA, and DHA) data were collected from Pétursdóttir et
al. (2008a, 2008b), Würzberg et al. (2011a, 2011b, 2011c); Lewis (1967), Howell et al. (2003),
Hudson et al. (2004), Økland et al. (2005), Drazen et al. (2008a, 2008b), Stowasser et al. (2009),
Murdukhovich et al. (2018), Parzanini et al. (2018a), Salvo et al. (2018), van Øevelen et al. (2018);
and Jeffreys et al. (2009) (Table S2).
**2.2 Statistical analysis**
Comparisons among multiple groups of deep-sea organisms were run through t-tests and oneway
analysis of variance (ANOVA). In particular, isotopic (i.e. $\delta^{15}N$, $\delta^{13}C$) and FA (i.e. ARA, EPA and
DHA) data were compared across organisms from different latitudes (i.e. tropical, temperate and
polar), habitats (i.e. pelagic, demersal, benthic), and collection depths (i.e. mesopelagic,
bathypelagic, meso-bathypelagic, abyssopelagic, bathyal, bathyal-abyssal, and abyssal) to detect
any significant differences. When the normality assumption was violated, Mann-Whitney rank sum



test, Kruskal-Wallis oneway ANOVA on ranks, and Dunn's method pairwise comparisons were
performed instead. In addition, multivariate statistics, i.e. principal coordinate analysis (PCO) and
permutational MANOVA (PERMANOVA) were used to study the variability in the isotopic and FA
composition of deep-water organisms across different latitudes, habitats, collection depths, and
phyla. In addition, a distance based linear model (DistLM) was run to assess which of these four
factors contributed the most to such a variability. PCO, PERMANOVA, and DistLM were run on
resemblance matrices, based on Euclidean distance for the isotopic data, and Bray-Curtis for the
FA data. Data were not normalized or transformed prior to analysis. Univariate statistics was
conducted using Sigmaplot 12.5, while PCO, PERMANOVA and DistLM were run through Primer
7.0 with the add-on package PERMANOVA+ (Clarke and Gorley, 2006).

## 2.3 Results

Analyses revealed both latitudinal and depth-related trends for isotopic and essential FA
composition. In particular, mean values (±SD) of $\delta^{15}N$ and $\delta^{13}C$ were significantly lower in deep-sea
fauna sampled at high latitudes than in that collected at low latitudes ($\delta^{15}N$, ANOVA on Ranks, $H$ =
69.1, $p \leq 0.001$; $\delta^{13}C$, ANOVA on Ranks, $H$ = 196.6, $p \leq 0.001$; Fig. 2). Conversely, no difference
was detected across latitudes in terms of ARA, but mean proportions (±SD) of EPA and DHA were
significantly greater at polar latitudes than at temperate and tropical areas (EPA, ANOVA on Ranks,
$H$ = 10.5, $p$ = 0.005; DHA, ANOVA on Ranks, $H$ = 52.0, $p \leq 0.001$; Fig. 3). Similarly, PERMANOVA
detected significant differences across latitudes in terms of both stable isotopes [Pseudo-F = 67.0,
$p(perm)$ = 0.0001] and essential FA [Pseudo-F =9.1, $p(perm)$ = 0.0001].

When deep-water species were analyzed separately, according to their habitat, the same

latitudinal trend in the isotopic composition were shown for deep-water benthic species ($\delta^{15}N$,
ANOVA on Ranks, $H$ = 64.5, $p \leq 0.001$; $\delta^{13}C$, ANOVA on Ranks, $H$ = 113.2, $p \leq 0.001$); whereas,
for demersal and pelagic species, only the $\delta^{13}C$ ratios were significantly lower at higher latitudes
(ANOVA on Ranks, $H$ = 97.9, $p \leq 0.001$; $t_{434}$ = -4.0, $p \leq 0.001$ ). PERMANOVA showed that the
isotopic composition of deep-sea animals was indeed statistically different across the three habitats



[Pseudo-F = 125.7, $p(perm)$ = 0.0001], and benthic and demersal species had higher stable N and
C isotope ratios than the pelagic counterparts ($p < 0.05$). Conversely, only benthic and pelagic
species revealed a latitudinal gradient in their essential FA composition (EPA, ANOVA on Ranks, $H$
= 10.2, $p$ = 0.006; DHA, ANOVA on Ranks, $H$ = 35.5, $p \leq 0.001$, for benthic species; EPA, ANOVA,
$H$ = 6.4, $p$ = 0.011). In this regard, pelagic, demersal, and benthic taxa had a different essential FA
composition (ARA, ANOVA on Ranks, $H$ = 35.0, $p \leq 0.001$; EPA, ANOVA on Ranks, $H$ = 12.5, $p$ =
0.002; DHA, ANOVA on Ranks, $H$ = 70.8, $p \leq 0.001$). Benthic species had the highest proportions
of ARA and EPA ($p < 0.05$), while demersal species had the highest levels of DHA, although similar
to those of pelagic species.

While mean values of both stable N and C isotope ratios increased with depth ($\delta^{15}$N,

ANOVA on Ranks, $H$ = 116.1, $p \leq 0.001$; $\delta^{13}$C, ANOVA on Ranks, $H$ = 122.7, $p \leq 0.001$),
proportions of EPA decreased along the bathymetric gradient for pelagic species (ANOVA on
Ranks, $H$ = 12.3, $p$ = 0.002). In addition, for benthic and demersal fauna, levels of $\delta^{15}$N, $\delta^{13}$C, and
ARA increased for benthic and demersal organisms with increasing depth ($\delta^{15}$N, ANOVA on Ranks,
$H$ = 84.7, $p \leq 0.001$; $\delta^{13}$C, ANOVA on Ranks, $H$ = 105.0, $p \leq 0.001$; ARA, ANOVA on Ranks, $H$ =
22.8, $p \leq 0.001$). PERMANOVA revealed significant differences in the isotopic [Pseudo-F = 89.5,
$p(perm)$ = 0.0001] and essential FA composition [Pseudo-F = 7.3, $p(perm)$ = 0.0001] across
collection depths.

Among the four variables considered (i.e. latitude, habitat, collection depth, and phylum),

analyses revealed that 'habitat' and 'phylum' were the most important factors influencing the
variability the stable isotope (respectively 13 and 9%; DistLM, $adjusted\ R^2$ = 0.4) and FA
(respectively 8 and 12%; DistLM, $adjusted\ R^2$ = 0.3) composition of deep-water organisms (Fig. 4).
**2.4 Discussion**
The present analysis shows for the first time, the existence of a) latitudinal trends in both stable
isotope and essential FA composition of deep-sea organisms, with decreasing $\delta^{13}$C ratios and
increasing ω3 LC-PUFA towards the poles; b) global bathymetric trends in the isotopic composition



of deep-water fauna for which mean levels of $\delta^{15}N$, $\delta^{13}C$, and ARA increased with increasing depth.
In addition, it provides further evidence of the link, across latitudes and depth, between surface
primary production of the surface waters and the deep-water consumers. The present findings
generally align with reports of decreasing values of $\delta^{13}C$ in surface-waters plankton and POM
towards the polar regions, in both the southern and northern hemisphere (Sackett et al., 1965; Rau
et al., 1982; Francois et al., 1993), as well as of increasing POM isotopic ratios along a bathymetric
gradient (Altabet et al., 1999). They also agree with Colombo et al. (2016) who noticed that
proportions of ω3 LC-PUFA were higher in marine organisms from polar and temperate regions in
comparison to tropical regions, and with Parzanini et al. (2018a) who detected increasing
proportions of ARA along a slope area in the deep Northwest Atlantic.

Water temperature, in combination with other abiotic (e.g. oceanographic and

biogeochemical processes, nutrient supply) and biological factors (e.g. species metabolism,
taxonomic composition of deep-water communities, microbial remineralization processes) seems to
play a role in these trends (Rau et al., 1982; Francois et al., 1993; Altabet et al., 1999; Colombo et
al., 2016). In particular, water temperature influences isotopic fractionation processes and, typically,
higher fractionation is associated with lower temperatures (Sackett et al., 1965). High fractionation
rates are also linked to the pronounced denitrification activities characterizing oligotrophic areas
such as observed in some areas of the tropics (Hetherington et al., 2017). This may explain the
higher $\delta^{15}N$ ratios of the deep-sea organisms from the tropical latitudes analyzed in this study.
Furthermore, water temperature affects membrane fluidity, and lower temperatures decrease the
fluidity of cell membrane (Parrish, 2013; Colombo et al., 2016). Thus, in order to maintain normal
membrane function and condition, i.e. health, ectotherms may counteract variations in water
temperature by readjusting their FA composition (Cossins and Lee, 1985; Parrish, 2013). For
example, larger proportions of long chain unsaturated FA (e.g. ARA, EPA) within the lipid bilayer
help increase membrane fluidity (Parrish 2013), as these molecules are characterized by a higher
flexibility (DeLong and Yayanos, 1985; Colombo et al., 2016).



Trends in the isotopic and FA composition of deep-sea organisms were also seen along a
depth gradient. As a proxy for water temperature as well as nutrient supply, depth may influence
biochemical composition of marine consumers (Parzanini et al., 2018a, 2018b). POM becomes
more isotopically enriched while sinking to deeper depth due to microbial degradation (Altabet et
al., 1999). Thus, the isotopic composition of deep-water organisms which feed on POM may vary
accordingly (Mintenbeck et al., 2007). In the present analysis, levels of ARA were globally higher at
deeper depths, similar to the study by Parzanini et al. (2018a), which may be due to i) a higher
reliance of deeper-dwelling organisms on the benthic-detrital trophic pathway; and/or ii) the need to
maintain membrane fluidity at low temperatures via increasing the unsaturation levels of membrane
phospholipids.
Finding latitudinal trends in the biochemical composition of deep-water organisms that
mirror results from shallow depths provides further evidence of the link between the two systems, in
that deep-sea benthic communities rely on POM sinking from the surface water as a primary food
source (Gage, 2003; Hudson et al., 2004). Close dependence of deep-sea food webs on near-
surface processes raises important concerns. According to the latest climate estimates, both air
and water temperatures have been rising, and continue to increase; and seawater pH has already
dropped by 0.1 units due to large $CO_2$ emissions, and is expected to decrease further (IPCC,
2017). Furthermore, models predict that increasing surface water temperature will favor
stratification, while reducing vertical mixing as well as enhancing variability in the transport of
primary production and energy (i.e. carbon) transport to the deep sea (Smith et al., 2009; Jones et
al., 2014; Sweetman et al., 2017). At the same time, deep-water benthic biomass is expected to
decrease due to the increasing variability in the food supply, which may in turn affect health and
functioning of benthic ecosystems, as well as global biogeochemical cycles (Jones et al., 2014).
Hixson and Arts (2016) showed that the FA composition of the six most common fresh- and salt-
water phytoplankton species responded to temperature and, specifically, that their ω3 PUFA levels
decreased with increasing temperature. Not only do ω3 PUFA, such as EPA and DHA, play an
important role in the response to temperature variations in aquatic systems, but they are also



essential nutrients and are highly required by aquatic organisms for optimal growth and health
(Parrish, 2009). A case in point, Rossoll et al. (2012) showed experimentally that growth and
reproduction of the copepod *Acartia tonsa* were severely compromised by the alteration of FA
content and composition of its primary food source, the diatom *Thalassiosira pseudonana*, exposed
to high $CO_2$ levels. The present investigation, therefore, suggests that changes in amounts and
composition of surface production could also result in changes in essential nutrients and
biomarkers in deep-sea benthic organisms that feed on it, with possible cascading effects
throughout deep-water food webs. Such variations may alter nutrient intake of deep-sea benthic
organisms, as well as trophodynamics; and they may also influence species' abilities to cope with
deep cold waters.

## 3 Conclusions

This investigation provides a first summary of the information available on deep-sea food webs
inferred by bulk stable isotope and FA analyses, providing guidance for future studies and a
glimpse at global-scale patterns in the biochemical composition of deep-water organisms. Food-
web tracers represent a powerful tool that can help elucidate the structure and dynamics of food
webs from shallow to deeper waters, and support management initiatives. However, this tool is
even more effective when combined with other techniques (e.g. gut content analysis), as each
method provides uniquely valuable data. When comparing studies, it emerges that there are
multiple sources of variations, whether biological, environmental, and/or analytical. Depending on
the scale of the investigation, these differences are more or less susceptible to biases, suggesting
that they have to be considered and acknowledged when attempting cross-comparisons even
though they may be contextually acceptable. The preliminary analysis conducted here detected
latitudinal and bathymetric trends in the isotopic and FA composition of deep-sea species. In light of
global climate change and the link between surface production and deep-sea communities,
changes in amounts and composition of surface production may influence the essential nutrient
intake (e.g. ω3 PUFA) of deep-water organisms. Because ω3 PUFA are involved in the response to



temperature variations in ectotherms, climate change may also affect the ability of these species to
cope with potential temperature shifts. However, more studies are required to help detect global
trends, especially in those areas that are still poorly understood (most deep-sea areas) or not yet
investigated (e.g. in the southern hemisphere). In addition, it is necessary to standardize analytical
methods to limit the influence and compensate for natural variability.
**Table S1. Dataset applied to analyze trends in the isotopic composition of deep-sea**
**animals.**
**Table S2. Dataset applied to analyze trends in the essential FA composition of deep-sea**
**animals.**
**Author contribution**
All the authors contributed to the manuscript conceptualization and methodology. CP was
responsible of data curation, formal analysis, investigation, and in writing the original draft of the
manuscript. CCP, JH, and AM reviewed and edited the draft. Lastly, CCP and AM provided
supervision, as well as funds to this project.
**Competing interests**
The authors declare that they have no conflict of interest.
**Acknowledgements**
The authors acknowledge the Natural Science and Engineering Research Council of Canada
(NSERC) Discovery Grant (Grant no. 311406 to A. Mercier and 105379 to C.C. Parrish) and
Canada Foundation for Innovation (CFI) Leaders Opportunity Fund (Grant no. 11231 to A. Mercier)
for funding. The authors also want to thank Jeff Drazen, Paul Snelgrove, Patrick Gagnon, E.



Montgomery and K. Bøe for providing ideas in the development and improvement of this
manuscript.



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



# Tables

**Table 1** Comparison outlining the major strengths and drawbacks of gut content, stable isotope, and FA analysis.

| Gut content analysis | Stable isotope analysis | FA analysis |
|---|---|---|
| Direct evidence of diet | Indirect evidence of diet (assumption validation required) | Indirect evidence of diet (assumption validation required) |
| Snap shot of the most recent meal | Integrative over time | Integrative over time |
| Small sample sizes lower representativity of diet | Small sample sizes may lower representativity of diet | Small sample sizes may lower representativity of diet |
| Inter-individual variability can only be accounted for with appropriate sample size | Inter-individual variability minimized due to integrative nature | Inter-individual variability likely but minimized due to integrative nature |
| Temporal variability can only be accounted for with appropriate sample size | Temporal variability minimized due to integrative nature | Temporal variability minimized due to integrative nature |
| Partly dependent on sex in cases where there are dietary differences between sexes | Partly dependent on sex in cases where there are dietary differences between sexes | Partly dependent on sex in cases where there are dietary differences between sexes |
| May be sensitive to body size (e.g. onthogenetic dietary changes) | May be sensitive to body size, whether or not size influences diet | Dependent body size if size affects diet |
| Species with large stomachs and slow digestion rates are easier to study | Applies to all species, but requires enough material (see below) | Applies to all species, but requires enough material (see below) |
| The analysis cannot be carried out with empty stomachs | Independent of stomach fullness | Independent of stomach fullness |
| Digestion rates may bias contents recovered | Independent of digestion process | Independent of digestion process |
| Small specimens with small stomachs are more difficult to study | Small specimens may have to be pooled, guts included | Small specimens may have to be pooled, guts included |
| Only gut content is analyzed | Typically applied to target tissues | Typically applied to target tissues |
| Interpretation is relatively easy, and the evidence obtained cannot be misinterpreted, taxonomically speaking | Data interpretation is complex (post-analysis mathematical corrections are often applied) | Data interpretation is complex (linked to FA biomarkers as food tracers) |
| Long processing time | Relatively short processing time | Relatively short processing time |
| Little instrumentation, low cost (unless high resolution scopes are used) | Medium tech, med/high cost | Medium tech, med/high cost |






**Table 2** Sources of variations across studies, distinguished by type (i.e. biological, environmental, analytical).

| Biological | Analytical | Environmental |
|---|---|---|
| Taxonomy | Sample gear | Depth |
| Sex | Sample storage | Season |
| Age | Sample treatment (e.g. Acidification of organisms containing carbonatic anatomical elements; Lipid removal; urea removal) | Primary productivity levels at surface |
| Size | Mathematical correction (i.e. whether applied and which one) | Latitude |
| Feeding habits | Tissue type | Temperature |
| General physiological condition | | Ocean region |





**Table 3** List of trophic ecology studies in deep-sea systems, that have been carried out using
stable isotopes (bulk) and lipids (including FA) as food-web tracers. Reference, method(s) applied,
latitude, sampling depth, ocean region, and taxa analyzed are reported for each study. Polar
latitudes include investigations between 60 - 90° N/S, whereas temperate and tropical latitudes
represent studies carried out within 0 - 30° N and 30 - 60° N, respectively. References are ordered
according to sampling depth(s).

| References | Method(s) | Latitude | Depth (m) | Ocean region | Taxa analyzed |
|---|---|---|---|---|---|
| Mintenbeck et al. 2007 | Stable isotopes | Polar | 50-1600 | Weddell Sea (Antarctic) | Benthic bryozoans, cnidarians, crustaceans, echinoderms, echiurans, mollusks, sponges, sipuncules, and tunicates |
| van Oevelen et al. 2018 | Stable isotopes, Lipids | Polar/Temperate | 270-850 | Trænadjupet Trough (Norwegian continental shelf), Belgica Mounds (Porcupine Seabight) | Cold-water coral communities |
| Würzberg et al. 2011a | Lipids | Polar | 600-5337 | Weddell Sea (Antarctic) | Shelf and deep-sea peracarid crustaceans + foraminiferans |
| Würzberg et al. 2011b | Lipids, Gut contents | Polar | 600-2150 | Weddell Sea (Antarctic) | Demersal fish |
| Würzberg et al. 2011c | Lipids | Polar | 600-5337 | Weddell Sea (Antarctic) | Shelf and deep-sea polychaetes |
| Iken et al. 2005 | Stable isotopes | Polar | 800-2082 | High Arctic Canadian Basin | Benthic cnidarians, crustaceans, echinoderms, echiurans, mollusks, and polychaetes; pelagic crustaceans |
| Pétursdóttir et al. 2008a | Stable isotopes, Lipids | Polar | 1000-2000 | Reykjanes Ridge (North Atlantic) | Mesopelagic crustaceans and fish |
| Pétursdóttir et al. 2008b | Stable isotopes, Lipids | Polar | 1000-2001 | Reykjanes Ridge (North Atlantic) | Mesopelagic crustaceans and fish |
| Bergmann et al. 2009 | Stable isotopes | Polar | 1300-5600 | HAUSGARTEN observatory, west Svalbard (Arctic) | Benthic cnidarians, crustaceans, echiurans, echinoderms, mollusks, nemertean worms, polychaetes, priapulids, sponges, and tunicates; Demersal fish |
| Valls et al. 2014a | Stable isotopes | Temperate | 40-400 | Balearic Basin (western Mediterranean) | Mesopelagic fish and zooplankton |
| Sherwood et al. 2008 | Stable isotopes | Temperate | 47-1433 | Northwest Atlantic | Cold-water corals |
| Hamoutene et al. 2008* | Lipids | Temperate | 50-1500 | Cape Chidley, and southern Grand Bank (Northwest Atlantic) | Cold-water corals |
| Boyle et al. 2012 | Stable isotopes, Gut contents | Temperate | 55-1280 | eastern North Pacific | Benthic cnidarians, crustaceans, echinoderms, and mollusks; polychaetes; demersal fish |





| | | | | | |
|---|---|---|---|---|---|
| Polunin et al. 2001 | Stable isotopes | Temperate | 200-1800 | Balearic Basin (western Mediterranean) | Demersal fish |
| Valls et al. 2014b | Stable isotopes | Temperate | 250-850 | Balearic Basin (western Mediterranean) | Hyperbenthic echinoderms and hyperbenthic/pelagic crustaceans, elasmobranchs and mollusks |
| Gale et al. 2013 | Stable isotopes, Gut contents | Temperate | 258-1418 | Northwest Atlantic | Echinoderms |
| Carlier et al. 2009 | Stable isotopes | Temperate | 300-1100 | Ionian Sea (central Mediterranean) | Cold-water coral community |
| Parzanini et al. 2018a | Stable isotopes, Lipids, Elemental | Temperate | 310-1413 | Northwest Atlantic | Slope cnidarians, crustaceans, echinoderms, fish, mollusks, sponges and tunicates |
| Parzanini et al. 2018b | Lipids | Temperate | 310-1413 | Northwest Atlantic | Slope cnidarians, crustaceans, echinoderms, fish, mollusks, sponges and tunicates |
| Parzanini et al. 2017 | Stable isotopes, Gut contents, Morphometrics | Temperate | 310-1413 | Northwest Atlantic | Pelagic and demersal fish |
| Madurell et al. 2008 | Stable isotopes | Temperate | 350-780 | Balearic Basin (western Mediterranean) | Suprabenthic crustaceans and fish |
| Kopp et al. 2018 | Stable isotopes | Temperate | 415-516 | Celtic Sea (Northeast Atlantic) | Epifaunal crustaceans, mollusks, and fish |
| Papiol et al. 2013 | Stable isotopes | Temperate | 423-1175 | Balearic Basin (western Mediterranean) | Benthopelagic crustaceans |
| Fanelli et al. 2013 | Stable isotopes | Temperate | 445-2198 | Balearic Basin (western Mediterranean) | Slope crustaceans and mollusks |
| Økland et al. 2004 | Lipids | Temperate | 500-1600 | Porcupine Bank and western continental slope (Northeast Atlantic) | Demersal fish |
| Trueman et al. 2014 | Stable isotopes | Temperate | 500-1500 | Hatton Bank (Northeast Atlantic) | Demersal fish |
| Kharlamenko et al. 2013 | Stable isotopes, Lipids | Temperate | 500-1600 | Sea of Japan | Echinoderms and mollusks |
| Preciado et al. 2017 | Stable isotopes, Gut contents | Temperate | 625-1800 | Galicia Bank (Northeast Atlantic) | Demersal fish and pelagic/demersal crustaceans |
| Fanelli et al. 2009 | Stable isotopes | Temperate | 650-780 | Algerian Basin (western Mediterranean) | Mesopelagic crustaceans and fish; benthic crustaceans |
| Fanelli et al. 2011a | Stable isotopes, Gut contents | Temperate | 650-800 | Balearic Basin (western Mediterranean) | Zooplankton and micronekton |
| Fanelli et al. 2011b | Stable isotopes | Temperate | 650-1000 | Balearic Basin (western Mediterranean) | Epibenthic/infaunal nemertin worms, polychaetes, sipuncules, mollusks, crustaceans, echinoderms |
| Salvo et al. 2017 | Lipids | Temperate | 770-1370 | Northwest Atlantic | Cold water corals |
| Stowasser et al. 2009 | Stable isotopes, Lipids, Gut contents | Temperate | 785-4814 | Porcupine Seabight and Abyssal Plain (Northeast Atlantic) | Moridae and Macrouridae fish |
| Hudson et al. 2004 | Lipids | Temperate | 800-4850 | Porcupine Seabight and Abyssal Plain (Northeast Atlantic) | Holoturoids |
| Howell et al. 2003 | Lipids | Temperate | 1053-4840 | Porcupine Abyssal Plain (Northeast Atlantic) | Asteroids |





| Tecchio et al. 2013 | Stable isotopes | Temperate | 1200-3000 | Mediterranean Sea (western + central + eastern) | Zooplankton |
|---|---|---|---|---|---|
| Reid et al. 2012 | Stable isotopes | Temperate | 2400-2750 | Mid-Atlantic Ridge (North Atlantic) | Benthic cnidarians, crustaceans, echinoderms, |
| Reid et al. 2013 | Stable isotopes | Temperate | 2404-2718 | Mid-Atlantic Ridge (North Atlantic) | Deep-sea fish |
| Mordukhovich et al. 2018 | Lipids | Temperate | 3352-4722 | Sea of Okhotsk and Pacific Ocean | Deep-sea nematodes |
| Drazen et al. 2008a | Lipids | Temperate | 4100 | eastern North Pacific | Ophiuroids and holoturoids |
| Drazen et al. 2008b | Lipids | Temperate | 4100 | eastern North Pacific | Cnidarians, polychaetes and crustaceans, demersal and pelagic crustaceans and fish |
| Drazen et al. 2008c* | Stable isotopes, Gut contents | Temperate | 4100 | eastern North Pacific | Macrourid fish |
| Drazen et al. 2009 | Lipids | Temperate | 4100 | eastern North Pacific | Macrourid fish and cephalopods |
| Iken et al. 2001 | Stable isotopes | Temperate | 4840 | Porcupine Abyssal Plain (Northeast Atlantic) | Demersal/Benthic cnidarians, crustaceans, echinoderms, echiurans, fish, mollusks, nematodes, polychaetas, sipuncules, and tunicates |
| Lewis, 1967 | Lipids | Tropical | 0-4000 | Off San Diego and Baja California (eastern Pacific) | Demersal and pelagic crustaceans and fish |
| Jeffreys et al. 2009 | Stable isotopes, Lipids | Tropical | 140-1400 | Arabian Sea | Crustaceans, cnidarians, and echinoderms |
| Churchill et al. 2015 | Stable isotopes, Gut contents | Tropical | 250-1200 | south-central Gulf of Mexico, off Florida to Louisiana (western Atlantic) | Elasmobranchs |
| Shipley et al. 2017 | Stable isotopes | Tropical/Polar | 472-1024 | Exuma Sound (The Bahamas), Lancaster Sound (Canadian Arctic) | Elasmobranchs |

*The study was excluded from analyses because it did not meet the criteria outlined in Sect. 2.1.1 or did not include any data.




## Figures caption

**Fig. 1.** Deep-sea biomarker studies in the world ocean. Symbols indicate where the studies listed in Table 2 have been carried out. In detail, red circles represent those investigations that have used stable isotopes as food web tracers; whereas yellow squares and green diamonds indicate those which used lipids and a combination of SIA and FA analysis, respectively.

**Fig. 2.** Stable N and C isotopic composition of deep-sea animals across latitudes. Mean values of $\delta^{15}N$ (blue circles above) and $\delta^{13}C$ (orange circles below) (‰) measured in deep-sea organisms across polar, temperate, and tropical latitudes. Bars represent standard deviation (n = 33 – 1479), and a letter code indicates significant differences ($p < 0.05$) across latitudes.

**Fig. 3.** Essential FA composition of deep-sea animals across latitudes. Mean proportions of essential FA measured in the tissues of deep-sea animals from polar (blue bars), temperate (orange diagonal striped bars), and tropical (green vertical striped bars) latitudes. Bars represent standard deviation (n = 7 – 212), and a letter code indicates significant differences ($p < 0.05$) across latitudes.

**Fig. 4.** Differences in terms of biochemical compositions among deep-sea animals from various habitats. Principal coordinate analysis plots representing differences in terms of isotopic (above) and essential FA composition (below) of deep-water species. In both cases, the variable 'habitat' resulted one of the most important factors, contributing 13 and 8% respectively to the variability in the biochemical composition of the deep-sea species.





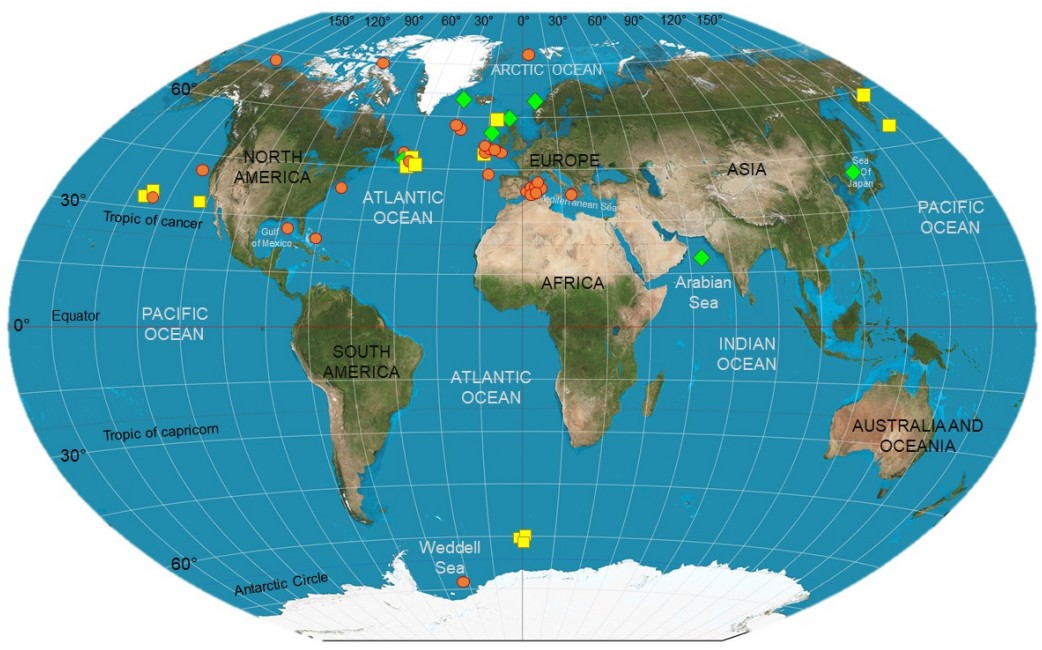

**Fig. 1.**



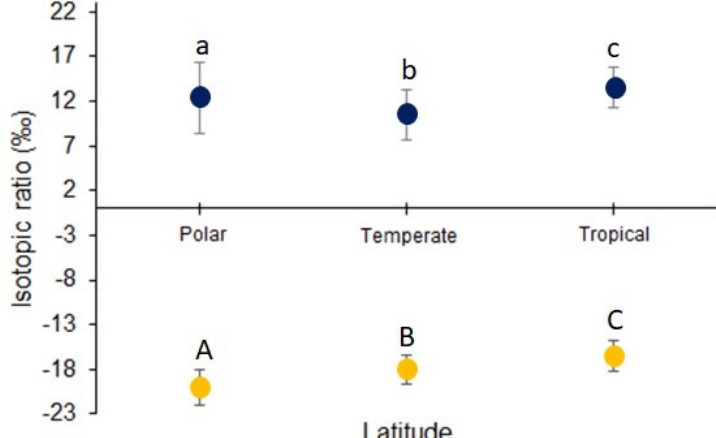

**Fig. 2.**





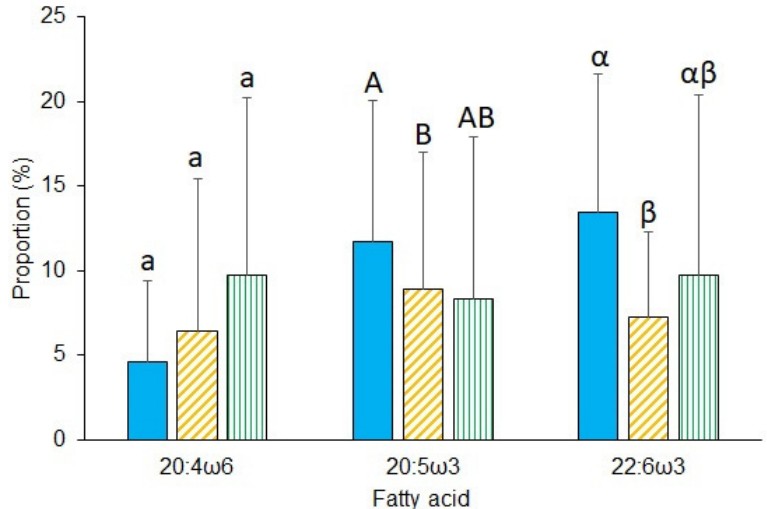

**Fig. 3.**



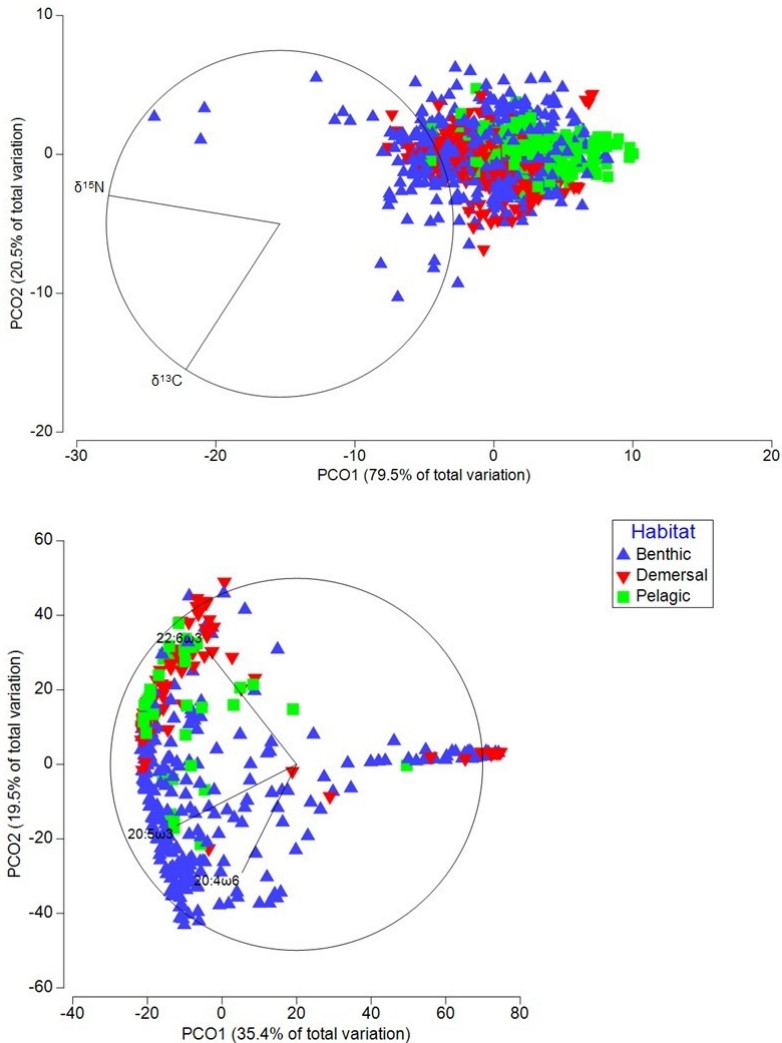


**Fig. 4.**

