# Peer review of "Reviews and syntheses: Insights into deep-sea food webs and global environmental gradients revealed by stable isotopes (δ15N, δ13C) and fatty acids trophic biomarkers"

_Biogeosciences, 2019_

## Referee Comment (RC1) · Anonymous Referee #1 · 5 Apr 2019

The manuscript "Insights into deep-sea food webs and global environmental gradients revealed by stable isotopes ($\delta$15N, $\delta$13C) and fatty acids trophic biomarkers" is the first attempt to summarize data on the use of stable isotopes and fatty acids as trophic markers for deep-sea ecosystems. The authors thoroughly analyze the practical aspects of application of these methods and suggest using the standardized methods in order to generate more reliable global predictions. Almost all currently available information was presented in this analysis. In general, the authors have shown convincingly the variations in fatty acid composition and isotope ratio of marine animals along the

latitudinal and bathymetric gradients. The main drawbacks of this study are not directly related to the authors' efforts and are associated with the scarcity of studies in the polar and, especially, tropical regions. Therefore, data on the tropical region can be considered as preliminary. I think that this manuscript is appropriate for publication by the "Biogeosciences" and should be of high interest to ecologists.
* * *

---

## Referee Comment (RC2) · Anonymous Referee #2 · 27 May 2019

The manuscript "Reviews and syntheses: Insights into deep-sea food webs and global environmental gradients revealed by stable isotopes (15N, 13C) and fatty acids trophic biomarkers" is an important contribution to the field of trophic ecology of deep-sea ecosystems. I like table 1, where the authors compared in detail the advantages and disadvantages of gut content, SI and FA analyses, and I highly appreciate the attempt to assess potential latitudinal and bathymetric trends in SI and FA.

I know that the dataset is very sparse, but apparently the keywords, the authors used, did not identify all literature published about this topic. I therefore listed several papers that should be included into the data analysis to increase its explanatory power. Gontikaki et al. 2011 (Deep-Sea Research I), Jeffreys et al. 2013 (Plos One) (mainly a tracer study, but it also includes natural abundance SI data), Jeffreys et al. 2015 (Biogeosciences), Kiyashko et al. 2014 (MEPS), Levin et al. 1999 (MEPS) (tracer study, but also natural abundance SI data), Lin et al. 2014 (MEPS), Mincks et al. 2008 (Deep-Sea Research II), Moens et al. 2007 (Polar Biology), Quiroga et al.2014 (MEPS), Sweetman & Witte 2008 (Deep-Sea Research I) (tracer study, but also natural abundance SI data), Veit-Köhler et al. 2013 (Progress in Oceanography).

There are likely more papers published, but these were the ones that came to my mind. Since the selection process for these papers does not follow the procedure described in the manuscript, the authors could include them under the term 'additional sources'.

I also miss information about the geological feature, i.e., whether samples were taken in canyons, at open slopes, in plains, etc. I assume that especially in canyons the SI composition of detritus that reaches the seafloor will be very different from plains at similar depth due to the faster transport of detritus down the canyon. This factor should also be investigated in the statistical analysis.

Minor comments are: The authors could mention earlier in the study that they explicitly excluded chemosynthetic studies. I know that it is mentioned in the MM section, but when I started reading the manuscript, I quickly went to the supplement to see which studies were included and I missed the chemosynthetic studies. Of course, this is absolutely related to the way I read the paper, but I could imagine that I am not the only one and that other readers would also like to know already in the abstract (or at the end of the introduction), that chemosynthetic studies were not included.

Table 1: The authors stated that the interpretation of gut content analysis is relatively easily, and that prey items cannot be taxonomically misidentified. I disagree here, as I think that it depends on the grade of digestion: Strongly digested prey items might not be identifiable.

Minor technical things in table 1: There is a 'may' missing in line 3 of gut content analysis. It should be 'Small sample sizes may lower representativity of diet'.

Please spell out tech and med. Do you mean technology or technique, methods, or something else?

Fig. 2 and 3: The author reported the sample size as n = 33-1470 and n = 7-212, respectively. I suggest reporting the sample size per latitude instead of a range. This helps the reader to interpret the results and see where the data are specifically sparse.

---

## Author Response (AR1)

**Response to Anonymous Reviewers**

**Reviewer#1**

COMMENT: The manuscript "Insights into deep-sea food webs and global environmental gradients revealed by stable isotopes ($\delta^{15}$N, $\delta^{13}$C) and fatty acids trophic biomarkers" is the first attempt to summarize data on the use of stable isotopes and fatty acids as trophic markers for deep-sea ecosystems. The authors thoroughly analyze the practical aspects of application of these methods and suggest using the standardized methods in order to generate more reliable global predictions. Almost all currently available information was presented in this analysis. In general, the authors have shown convincingly the variations in fatty acid composition and isotope ratio of marine animals along the latitudinal and bathymetric gradients. The main drawbacks of this study are not directly related to the authors' efforts and are associated with the scarcity of studies in the polar and, especially, tropical regions. Therefore, data on the tropical region can be considered as preliminary. I think that this manuscript is appropriate for publication by the "Biogeosciences" and should be of high interest to ecologists.

REPLY: Many thanks to the anonymous referee #1 for the positive feedback suggesting the publication of this study. Specifically, my co-authors and I are glad the referee #1 was able to recognize the value of our study, while understanding its caveats derived from the limited number of investigations in certain areas of the globe.

**Reviewer#2**

COMMENT 1: The manuscript "Reviews and syntheses: Insights into deep-sea food webs and global environmental gradients revealed by stable isotopes (15N, 13C) and fatty acids trophic biomarkers" is an important contribution to the field of trophic ecology of deep-sea ecosystems. I like table 1, where the authors compared in detail the advantages and disadvantages of gut content, SI and FA analyses, and I highly appreciate the attempt to assess potential latitudinal and bathymetric trends in SI and FA.

REPLY: My coauthors and I would like to thank you for the insightful feedback and suggestions.

COMMENT 2: I know that the dataset is very sparse, but apparently the keywords, the authors used, did not identify all literature published about this topic. I therefore listed several papers that should be included into the data analysis to increase its explanatory power. Gontikaki et al. 2011 (Deep-Sea Research I), Jeffreys et al. 2013 (Plos One) (mainly a tracer study, but it also includes natural abundance SI data), Jeffreys et al. 2015 (Biogeosciences), Kiyashko et al. 2014 (MEPS), Levin et al. 1999 (MEPS) (tracer study, but also natural abundance SI data), Lin et al. 2014 (MEPS), Mincks et al. 2008 (Deep-Sea Research II), Moens et al. 2007 (Polar Biology), Quiroga et al.2014 (MEPS), Sweetman & Witte 2008 (Deep-Sea Research I) (tracer study, but also natural abundance SI data), Veit-Köhler et al. 2013 (Progress in Oceanography). There are likely more papers published, but these were the ones that came to my mind. Since the selection process for these papers does not follow the procedure described in the manuscript, the authors could include them under the term 'additional sources'.

REPLY: Thank you for providing new references that our previous search did not find. All the suggestions provided were carefully examined. While we used data from a number of them, some studies were not considered relevant to our analysis as they were either experimental (e.g. Levin et al. 1999, Gontikaki et al. 2011) or dealt with meiofauna (e.g. Veit-Köhler et al 2013)/foraminifera (Jeffreys et al. 2015). In addition, a few new studies were found after conducting a final search; therefore, 6 more studies were added to our data set for analysis. Results from updated statistical analyses are presented, along with updated version of Fig. 1, 2, and 3; nonetheless, conclusions have remained unchanged.

COMMENT 3: I also miss information about the geological feature, i.e., whether samples were taken in canyons, at open slopes, in plains, etc. I assume that especially in canyons the SI composition of detritus that reaches the seafloor will be very different from plains at similar depth due to the faster transport of detritus down the canyon. This factor should also be investigated in the statistical analysis.

REPLY: While this comment was pertinent and valuable, we eventually decided not to run new statistical analyses considering varying geological features because our primary goal was to assess variations at a global spatial scale, as it has been stated a few times throughout the text (e.g. see lines 48, 334, 367 of edited manuscript). Including 'Geographical feature" as a variable would have narrowed down our focus. Indeed, for the same aforementioned reason, we had originally excluded other variables, whether environmental (e.g. season) or biological (e.g. size, trophic group), from our assessment. Furthermore, by including the 'Geological feature' parameter to our analysis, all the studies conducted in pelagic habitats would have been automatically excluded, thus reducing sample size. In addition, the records would have been biased towards areas more commonly represented (e.g. slope) than others (e.g. ridges, trenches). Nonetheless, we added this additional source of variation (Environmental) in Table 2.

COMMENT 4: The authors could mention earlier in the study that they explicitly excluded chemosynthetic studies. I know that it is mentioned in the MM section, but when I started reading the manuscript, I quickly went to the supplement to see which studies were included and I missed the chemosynthetic studies. Of course, this is absolutely related to the way I read the paper, but I could imagine that I am not the only one and that other readers would also like to know already in the abstract (or at the end of the introduction), that chemosynthetic studies were not included.

REPLY: Agreed. This information has been added in the Abstract (line 7 of edited manuscript), Introduction (line 47), as well as the Materials and Methods (line 339) and Conclusions (lines 514-515).

COMMENT 5: Table 1: The authors stated that the interpretation of gut content analysis is relatively easily, and that prey items cannot be taxonomically misidentified. I disagree here, as I think that it depends on the grade of digestion: Strongly digested prey items might not be identifiable.

REPLY: We completely agree with this comment. Depending on the digestion level, it is more or less possible to identify a prey item. The sentence in the Table has been adjusted accordingly.

COMMENT 6: Minor technical things in table 1: There is a 'may' missing in line 3 of gut content analysis. It should be 'Small sample sizes may lower representativity of diet'.

REPLY: Added.

COMMENT 7: Please spell out tech and med. Do you mean technology or technique, methods, or something else?

REPLY: Both terms have been spelled out accordingly in Table 1.

COMMENT 8: Fig. 2 and 3: The author reported the sample size as n = 33-1470 and n = 7-212, respectively. I suggest reporting the sample size per latitude instead of a range. This helps the reader to interpret the results and see where the data are specifically sparse.

REPLY: Samples sizes in Fig. 2 and 3 have now been reported by latitude, as suggested.

[revised manuscript text omitted]